# Introducing Uncertainty in Risk Calculation along Roads Using a Simple Stochastic Approach

**Michel Jaboyedoff** [1,*], **Tiggi Choanji** [1,2], **Marc-Henri Derron** [1], **Li Fei** [1], **Amalia Gutierrez** [1], **Lidia Loiotine** [1,3], **François Noel** [1,4], **Chunwei Sun** [1,5], **Emmanuel Wyser** [1] and **Charlotte Wolff** [1]

1. Risk-Group, Institute of Earth Sciences, University of Lausanne, GEOPOLIS—3793, CH-1015 Lausanne, Switzerland; Tiggi.Choanji@unil.ch or tiggich@eng.uir.ac.id (T.C.); Marc-Henri.Derron@unil.ch (M.-H.D.); Li.Fei@unil.ch (L.F.); Carlota.Gutierrez@unil.ch (A.G.); Lidia.Loiotine@unil.ch or lidia.loiotine@uniba.it (L.L.); Francois.Noel@NGU.NO (F.N.); sunchunwei@my.swjtu.edu.cn (C.S.); Emmanuel.Wyser@unil.ch (E.W.); Charlotte.Wolff@unil.ch (C.W.)
2. Faculty of Engineering, Department of Geological Engineering, Universitas Islam Riau, Jl. Kaharuddin Nasution 113, Riau 28284, Indonesia
3. Dipartimento di Scienze della Terra e Geoambientali, Università Degli Studi di Bari ALDO MORO, via E. Orabona 4, 70125 Bari, Italy
4. NGU, Leiv Eirikssons vei 39, 7040 Trondheim, Norway
5. Faculty of Geosciences and Environmental Engineering, Southwest Jiaotong University, Chengdu 611756, China
* Correspondence: Michel.Jaboyedoff@unil.ch

**Abstract:** Based on a previous risk calculation study conducted along a road corridor, risk is recalculated using a stochastic simulation by introducing variability into most of the parameters in the risk equation. This leads to an exceedance curve comparable to those of catastrophe models. This approach introduces uncertainty into the risk calculation in a simple way, and it can be used for poorly documented cases to compensate for a lack of data. This approach tends to minimize risk or question risk calculations.

**Keywords:** landslide; rockfall; risk; stochastic; uncertainty; transportation corridors

## 1. Introduction

Several authors have used power-laws to assess hazards as functions of the volume or area of instability [1–4] or risk [5]. Volumes are often used as quantifications of the magnitudes of landslides. The frequency of failure of a volume greater than a given volume *Vol* [3] for a given region and several observations $N_0$ during a period $\Delta t$ is given by

$$\lambda(v \geq Vol) = \frac{N_0}{\Delta t}\left(\frac{Vol}{V_0}\right)^{-b} = a\,Vol^{-b} \tag{1}$$

In general, an analysis is based on the following conceptual formula (modified from [6]):

$$R = \lambda_r \times f_r \times PS \times Pp \times Exp \times E \times V \tag{2}$$

where $\lambda_r$ is the temporal frequency of rupture for a given period within a given perimeter and $f_r$ is the probability of rupture associated with a given magnitude or volume (here, $\lambda = \lambda_r \times f_r$). *PS* is a spatial weight if the exact location is not known, *Pp* is the probability of propagation calculated from the rockfall source at a given location, *Exp* is the exposure, and *E* corresponds to the value or unit of the object at risk and *V* is its vulnerability.

One of the problems is that this formulation does not often explicitly incorporate uncertainty, even though [7] proposed this approach using the term "risk curve". Uncertainty has mainly been applied by introducing random variables into the calculation of the factor of safety [8,9]. Uncertainty can also be inserted by using first-order second-moment (FOSM)

methods, for which an objective function is chosen that is supposed to respect a Gaussian distribution, for example, the factor of safety, whose analytical expression is known; the variances of the variables should also be known [6,7,10]. Wang et al. (2014) [11] applied the FOSM technique for inserting uncertainty in the risk analysis of block falls potentially affecting a tourist area and showed that the one-sigma confidence interval varies from 48% to 132% of the mean value. Simulations of block trajectories can provide probabilities of exceedance as a function of impact energy on objects [12]. Macciotta et al., (2016) [13] showed that by inserting uncertainty into Monte Carlo simulations, the risk of rockfall on a section of railway track is reduced.

Here, the analysis carried out by [5] along a road section is taken up again and simplified by replacing some parameters with random variables and by using Monte Carlo simulations via MATLAB 2018b (see Supplementary Materials). The approach is comparable to that of [13], but the intention is to show that such an approach can particularly be applied when data are lacking; this is similar to the disaster model [14], which presents its results according to an exceedance curve with no particular constraints.

## 2. Model Data

Hungr et al. (1999) [5] used Equation (1) and provided a simple synthetic example of risk calculation along a stretch of roads in British Columbia; this is adapted to follow the calculations used in this chapter. On average, [5] calculated that $N_0 = 100$ events reach the road per year for volumes greater than $V_0 = 0.001$ m$^3$, and they are distributed according to a cumulative power with the observed $b$ equal to 0.434 and $a = N_0 \times V_0^b = 4.99$ (Figure 1):

$$\lambda(v \geq Vol) = \frac{100}{1\ year} \left( \frac{Vol}{0.001} \right)^{-0.434} = 4.99\ Vol^{-0.434} \tag{3}$$

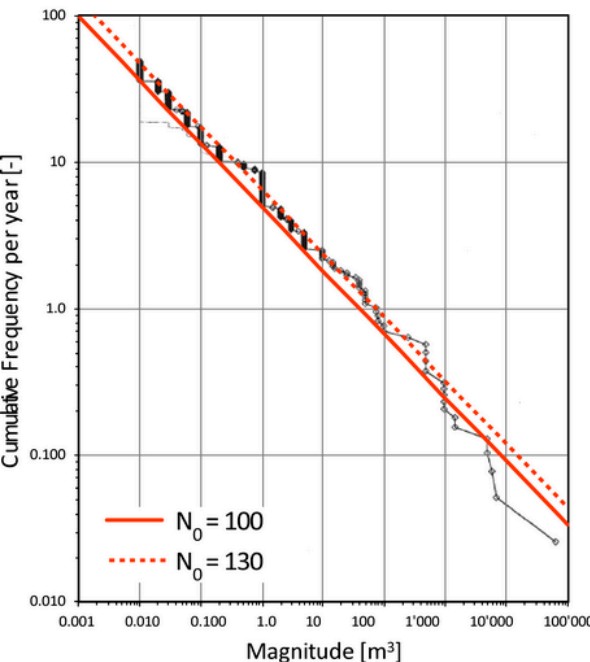

**Figure 1.** Cumulative frequency distribution as a function of the magnitude (volume) of 390 events along 75 km of Highway 99 in British Columbia; this includes the adjustment proposed by [5] for 100 events per year, as well as a second curve when this value is modified to 130 events per year (modified from [5]).

We obtain the frequency of each class of volumes, i.e., by calculating the difference between the values obtained for the two limits of the class via Equation (3). PS (see Equation (2)) is equal to 1 since it is known that it reaches the road section under consideration. The probability of propagation is relative to the location of the object. According to [5], as we have a two-way road, small volumes (<5 m$^3$) affect only one of the lanes, and these smaller volumes do not necessarily affect a car passing over them; however, for volumes above 100 m$^3$, the road section is fully covered by the rockfall over a width $D$ and $Pp = 1$. Exposure is calculated according to $D$, which increases roughly with the cubic root of the volume. The average vehicle length $L_v$ is 5.4 m, and 5000 vehicles travel per day. Here, only fatal accidents for at least one occupant are counted; therefore, vulnerability is equal to lethality, injuries are not considered, and E is implicitly set to 1. As an example, using the values chosen by [5] for the class of blocks from 0.1 to 1 m$^3$, we obtain (Table 1):

$$
\begin{aligned}
R\left(0.1-1\ m^3\right) &= \left(\lambda_r \times f_r\right) \times PS \times Pp \times Exp \times E \times V \\
&= 8.56 \times 1 \times 0.4 \times 0.0167 \times 1 \times 0.2 \\
&= 0.011\ fatal\ accidents\ per\ year
\end{aligned}
\tag{4}
$$

**Table 1.** Details of the risk calculations for different classes recalculated from [5], but with D = Vol$^{1/3}$ (see Supplementary Materials).

| Volume | $4.99 \times Vol^{-0.434}$ | $\lambda_r \times f_r$ | $D \sim Vol^{(1/3)}$ | $Exp$ | $Pp$ | $V$ | $H \times Pp \times Exp \times V$ | $1/R$ |
|---|---|---|---|---|---|---|---|---|
| (m$^3$) | (#/yr) | (#/yr) | (m) | (-) | (-) | (-) | (-) | (yr) |
| 0.001 | 100.000 | | | | | | | |
| 0.010 | 36.813 | 63.187 | 0.2 | 0.0146 | 0.1 | 0.05 | 0.005 | 217.0 |
| 0.100 | 13.552 | 23.261 | 0.5 | 0.0154 | 0.2 | 0.1 | 0.007 | 139.9 |
| 1.0 | 4.989 | 8.563 | 1 | 0.0167 | 0.4 | 0.2 | 0.011 | 87.6 |
| 10 | 1.837 | 3.152 | 2 | 0.0193 | 0.6 | 0.5 | 0.018 | 54.9 |
| 100 | 0.676 | 1.160 | 5 | 0.0271 | 0.8 | 0.8 | 0.020 | 49.7 |
| 1000 | 0.249 | 0.427 | 10 | 0.0401 | 1.0 | 1.0 | 0.017 | 58.4 |
| 10,000 | 0.092 | 0.157 | 30 | 0.0922 | 1.0 | 1.0 | 0.014 | 69.0 |
| >10,000 | | 0.092 | 50 | 0.1443 | 1.0 | 1.0 | 0.013 | 75.7 |
| | | | | | | **Total** | 0.106 | 9.4 |

The exposure is recalculated according to [15]:

$$
Exp = N_v \frac{(L_v + D)}{v_v} = \frac{5000}{24} \frac{(5.4 + 1)}{80 \times 1000} = 0.0167
\tag{5}
$$

where $v_v$ is the speed of a vehicle and $N_v$ is the number of vehicles per year. The sum of all classes up to $10^5$ m$^3$ indicates an average annual frequency of fatal accidents of 0.106, i.e., approximately one accident every 10 years (Table 1). By using the upper bounds of the classes, the risk is increased when compared to that obtained through the use of the average of the classes. The following paragraph attempts to overcome this problem by introducing simulations, which allow uncertainty to be incorporated in the model, and the values of the vulnerability or probability of death and the probability of impact are modified according to functions instead of with the discrete sets of values used by [5] (Figure 2).

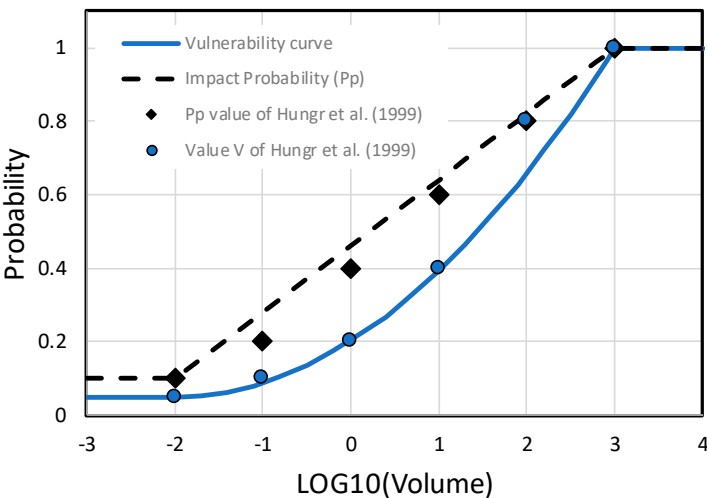

**Figure 2.** Model for the probability of impact or spread (Pp) and a vulnerability (V) curve created from data from [5] to make the functions continuous.

### 3. Introducing Uncertainty into Risk Calculation

Currently, the use of related uncertainty is increasingly required for risk management, and one of the means to obtain it is to use risk calculation simulations. This is presented through a previous example of risk calculation by modifying the procedure of [5]. The first step of the simulation is conducted according to the distribution of the volumes that will fall, and thus it is necessary to define the minimum and maximum annual frequencies corresponding to the maximum ($10^5$ m$^3$) and minimum ($10^{-3}$ m$^3$) volumes of the distribution function, respectively. Let $F_{max}$ = 4.99 × 0.001$^{0.434}$ = 100 and $F_{min}$ = 4.99 × 100,000$^{0.434}$ = 0.0337. Starting from the power-law cumulative distribution, it is quite easily inverted and thus, we can draw values at random in an equiprobable way between $F_{min}$ and $F_{max}$ such that the simulated frequency is given by

$$F_{sim} = F_{min} + rnd \times (F_{max} - F_{min}) \tag{6}$$

where *rnd* is a random variable varying from 0 to 1 according to a uniform distribution. Thus, the corresponding volume is

$$V_{sim} = \left(\frac{F_{sim}}{a}\right)^{-\frac{1}{b}} \tag{7}$$

This makes it possible to simulate a distribution of rockfall events per year. Instead of calculating by class, the calculation is performed for each of the 100 simulated volumes. Based on these simulations, it is possible to add distributions for several variables into the risk calculation. First, the number of events is, on average, 100 events per year; we can make the number of events into a random variable by using an inverse Poisson distribution, which allows us to simulate random values from a mean for discrete values. One million years are simulated (Figure 3).

In the example of [5], there were two estimated variables that are discrete: *Pp* and *V*. As mentioned earlier, the idea is to make them continuous via a linear fit for *Pp* and a second-degree polynomial for *V* from the log base 10 values of the volumes (*Vol*) (Figure 2):

$$Pp = 0.180 \, log_{10}(Vol) + 0.460 \tag{8}$$

$$V = 0.038 \, (log_{10}(Vol))^2 + 0.152 \, log_{10}(Vol) + 0.202 \tag{9}$$

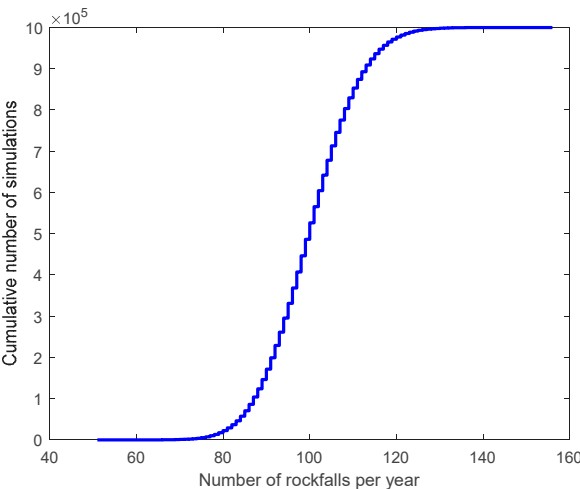

**Figure 3.** Cumulative distribution of the number of events per year for the $10^6$ simulations, based on a Poisson distribution using an average parameter of 100.

The value of $D$ is given by the cubic root of the volume. The last step is to add distribution functions to the other variables. For simplification, uniform distribution functions are used here, i.e., the values are equiprobable between two limits (Table 2). This applies to the variables related to the exposure: $D$, $v_v$, and $N_v$. We do not randomize $L_v$ because the length of the zone affecting the passengers is not easy to estimate and does not change much; the goal is also to be coherent with [5]. The limits are chosen based on so-called "expert knowledge" while assuming reasonable ranges for values centered on the average value.

**Table 2.** Limitations of uniform distributions of random variables.

| Variables | Units (Remarks) | Minimum | Maximum |
|---|---|---|---|
| Debris width $D$ | m | $D/2$ | $3D/2$ |
| Vehicle speed $v_v$ | km/h | 57.5 | 102.5 |
| Number of vehicles $N_v$ | Vehicles/day | 4500 | 5500 |
| Probability of impact or propagation at the vehicle location $Pp$ | (-) (Integrated in the calculation; one order of magnitude of volume variability) | $\log_{10}(V(d)) - 0.5$ | $\log_{10}(V(d)) + 0.5$ |
| Vulnerability $V$(lethality) | idem | idem | idem |

## 4. Results

The simulation program is executed first for 10,000 events with the same data as those in Table 1 [5], except for the continuous functions of $V$ and $Pp$; the annual frequency of accidents is 0. 0992, i.e., one fatal accident every 10 years. By simply adding the variabilities shown in (Table 2), for 10,000 simulations, we obtain 0.103 (1 accident every 9.7 years), which shows that for the selected distributions, the results converge comparably to the data in [5], even with a relatively small number of simulations.

By carrying out $10^6$ simulations for one year with a number of annual rockfalls distributed according to Figure 3, we obtain an average frequency of 0.059 events per year, i.e., one event every 16.8 years (Table 3). The median is 0.047, i.e., a longer time than that obtained by [5] separates the potential accidents. The fact that we are no longer working with classes reduces the average frequency (it is divided by almost half). The so-called exceedance curves indicate that there is a 95% chance that there are less than 45.6 years between two events (Figure 4). The probability of having an event within less than 7.3 years is 5%, which is not negligible.



**Table 3.** Characteristics of the exceedance curves in Figure 4 for the first two columns and for two other scenarios obtained by changing the number of occupants in the car and the average total number of rockfalls per year.

| Thresholds | Frequency | Return Period T [Year] | | | |
|---|---|---|---|---|---|
| **Case** | **A** | **A** | **B** | **C** | **D** |
| | **(events/year)** | **1 occ. $N_0 = 100$** | **1 occ. $N_0 = 130$** | **1–2 occ. $N_0 = 100$** | **1–2 occ. $N_0 = 130$** |
| **Average** | 0.059 | 16.8 | 13.0 | 11.2 | 8.6 |
| **Minimum (max. *T*)** | 0.010 | 103.3 | 80.9 | 75.4 | 48.6 |
| **97.50%** | 0.020 | 51.2 | 35.4 | 34.8 | 24.1 |
| **95%** | 0.022 | 45.6 | 31.8 | 31.0 | 21.6 |
| **Median** | 0.047 | 21.1 | 15.5 | 14.2 | 10.5 |
| **5%** | 0.137 | 7.3 | 6.0 | 4.8 | 3.9 |
| **2.5** | 0.165 | 6.1 | 5.1 | 3.9 | 3.3 |
| **Maximum (Min. *T*)** | 0.603 | 1.7 | 1.6 | 1.2 | 1.0 |

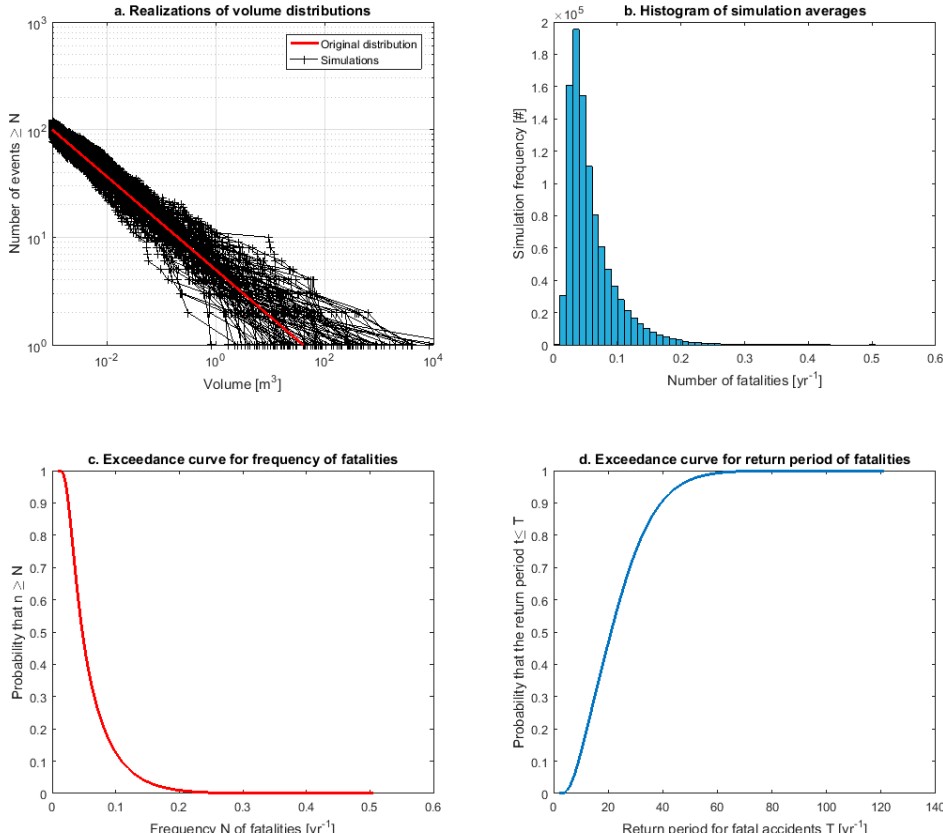

**Figure 4.** Simulation results. (**a**) 10 realizations of the volume distributions; (**b**) histogram of the simulated fatal accident frequencies; (**c**) exceedance curve or probability that the frequency is greater than a given value; (**d**) probability that the accident return period is smaller than a given value.

## 5. Discussion and Conclusions

The orders of magnitude are respected since [5] indicated that the return period of fatal accidents observed on Highway 99 between 1960 and 1996 was 12 years and 8 years from 1980 to 1996, as traffic increased. Here, the mean and median values are *T* = 17 and 21 years, respectively, and 95% of the simulated return periods are greater than 7.3 years, which is close to the observation. This result can be interpreted in different ways, either by using high probability thresholds or by modifying the distributions of the random

variables introduced, which are nevertheless symmetrical. Alternatively, the recently produced accident statistics, and an analysis of the accidents according to collisions must be questioned, as they could be added and halve the simulated return period.

By increasing the number of events per year to $N_0 = 130$, we also fit the data (Figure 1) by maximizing the frequency; the average return period is 12.9 years (median 15.3) (Table 3, case B). By using a random number of occupants (1 or 2), we obtain that T = 11.2 years (median 14.1; case C), and if both are used, the result is 8.6 years (median 10.4; case D). This shows that a reasonable hypothesis can lead to agreement with the observed data. It also shows that for models A to D, the exceedance probability of 5% of the return period ranges from 3.9 and 7.3 years. It is also noteworthy that the centered 95% confidence level ranges for the return period decrease with hazard and occupant increases; for cases A to D, the results are 51.2 to 6.1 years (range 45.1), 35.4 to 5.1 years (30.3), 34.8 to 3.9 years (30.9), and 24.1 to 3.3 years (20.8), respectively.

This approach makes it possible to add probabilities of realization to frequencies or return periods, and this is useful for decision-making. The above example permits us to analyze the sensitivity of risk calculations. Randomizing the original data from [5] minimizes the average risk because it enables the calculation values for all realizations and not just for classes, while at the same time providing elements for the quantification of uncertainties. Some assumptions concerning the distributions of the variables $D$, $v_v$, and $N_v$ are not simple, but it seems that reasonable choices provide reliable results. Haimes (2015) [16] also showed that performing the risk calculation using a probabilistic approach reduced the risk compared to the average value. This type of approach is likely to be developed in landslide risk assessments, such as those of propagation models, by also introducing variability. This is a way to introduce the catastrophe model [14] into landslide risk assessments.

The main objective of this study is to show that this kind of method can be applied easily by adding other random variables while using other distribution functions, such as the normal distribution when the variable distribution is assumed to be symmetrical and the standard deviation can be estimated; the log-normal distribution is used if large values present a large tail; the triangular distribution is well designed for expert knowledge because the minimum and maximum values are needed in addition to the most likely value [17]. In any case, the use of Poisson distributions is a valid approach when nothing is known about the integer values except the mean. This method becomes especially useful when the knowledge of the data is partial, meaning that it is possible to obtain an exceedance curve using expert input, as proposed by [13] and [11]. Such sensitivity studies should be used more often in the near future, but recommendations should be issued so that the results can be compared for risk management purposes.

**Supplementary Materials:** The following are available online at https://www.mdpi.com/2076-3263/11/3/143/s1, Figure S1: title, Table S1: title, Video S1: title.Code (Matlab 2018b): Simul_risk_rockfall_road.m, recalculation of table from [5]: S_Table_Hungr_et_al_1999.xlsx.

**Author Contributions:** The first authors proposed the method and wrote the computer code with the help of E.W., and the design of the study was set up during a workshop where all the authors contributed. All authors have read and agreed to the published version of the manuscript.

**Funding:** T.C. was funded by a grant from the Indonesia Endowment Fund for Education (LPDP), grant number 201905220214428, financial support for C.S. was sponsored by the China Scholarship Council grant number No. 201907000093., and C.W. was supported by a Canton of Ticino (Switzerland) project.

**Acknowledgments:** We thank two anonymous reviewers for their constructive comments and American Journal Expert for the improvement of the English language.

**Conflicts of Interest:** The authors declare no conflict of interest.

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
