# Peer review of "Introducing Uncertainty in Risk Calculation along Roads Using a Simple Stochastic Approach"

_geosciences, doi:10.3390/geosciences11030143_

Round 1
Reviewer 1 Report
Dear Authors,
The expression of risk using uncertainties, as this paper states, is for sure a very relevant issue for scientists and practitioners. This paper, deals with the incorporation of these uncertainties into the rockfall risk assessment with the aim to present the risk in terms of probability of exceedance-excess of a certain level of loss, compared to the average expected annual loss that is widely used in the literature during the last decades.
In my opinion, the probabilistic expression of risk is totally acceptable and useful, however it is not new. This way of expressing risk has been described beforehand as for example at the following article, p. 41, section: Suggested methods for quantitative landslide risk analysis
Corominas, J., van Westen, C., Frattini, P., Cascini, L., Malet, J. P., Fotopoulou, S., ... & Smith, J. T. (2014). Recommendations for the quantitative analysis of landslide risk. Bulletin of engineering geology and the environment, 73(2), 209-263.
Additionally, the risk expressed in terms of vulnerability curves, which show the probability of exceedance of a given degree of damage/loss in the same way as the risk is presented in this paper, is not new.
In any case, there are scarce applications of these probabilistic methods to the rockfall risk assessment, that is why I believe that this paper is of interest, however provided that a more extensive review of the already existing framework where the is expressed in terms of exceedance curves is included. This part needs to be stressed out.
Further comments:
In page 2, line 75 it is mentioned that the way of calculating the risk (e.g. using Equation 2 which provides the risk in terms of average annual loss) , and using average values for the frequency, the exposure etc… is conservative. This statement is not proved. Whether it is conservative or not will mostly depend on the distribution of the variables that enter into the equation, and whether the values used for the application of Equation 2, considering the shape and deviation of the distribution …So I think that this statement is not certain.
Page 4, line 93: I suggest stating Fmin and Fmax as being annual frequency instead of just frequency
Equation (7): Resulting from Equation 1, the exponent in Equation 7 should be -1/b and not 1/b. Additionally, for the value of the frequency that is used in Equation 7, according to the theoretical Equation (1), the volume for the real events could be Vsim or greater than Vsim, which means that the volume for the observed events could be greater than the volumes generated with this simulation. Can the authors check and provide data for the distribution of the random sample of events that give an insight on the accuracy of the distribution of the generated samples? I think this point needs clarification, in the methodology.
Page 4, lines 104-107 and Figure 3: The power law distribution was assumed to be valid for the size distribution of events, within a year. On what data are the authors based in order to make the assumption that the number of events also follows a power law distribution? This assumption needs further justification.
In page 4 line 104 it is stated that the number of samples for the Monte Carlo simulation is 10000. Again, could the authors explain more on the accuracy of the distribution of the random sample, compared to the input “theoretical” distribution? Is 10000 years enough to have an accurate sample?
Equations (8) and (9). Please explain what is the flight
Page 5: line 115. As stated by the authors assessing the distribution of the risk components is a challenge. If a uniform distribution is used, the procedure fails to take into consideration the conditional probabilities between the hazard, exposure and vulnerability components. Could you please explain if and how are those conditional probabilities taken into account using the proposed procedure?
Table 2: Please clarify what is meant with the phrase “integrated in the calculation from the integration of an order of magnitude of the volume”
Page 5, line 123. I think the term annual frequency should be used here instead of just frequency. Also, for the given example that the annual frequency is lower than 1, the annual probability could be taken as the annual frequency, given the relatively high return period of events. However, for higher frequencies (e.g. higher than 1), the annual probability of 1 or more events is not the same as the frequency and the probability can be calculated using the annual frequency by the binomial theorem. So I suggest either that you add this remark, or that you only use the term annual probability, which indeed is what is calculated from the Monte Carlo simulation.
Page 5, line 126. How would this number change if you would perform more simulations as for example 100.000? What is the accuracy of the method, using the selected sample size?
Page 5, line 126. I think that more than validity this example and its results show that for the selected distributions the results converge.
Page 6, line 137. Table 3: Concerning “excess supply curves”. I have some doubts on whether this term is applicable here as this is not the case of an element failure, where there is the issue of resistance over demand..if the risk if higher than 0, in a way there is always failure and the supply is negative. This is just a matter of terminology, but I see the supply more of a term that represents resistance than demand. Maybe just loss excess curves could be an alternative.
Figure 4b. It is not very clear to me what the number of results by classes here refers to.
Figure 4c. Shouldn´t it be n>N in the vertical axis?
Minor comments:
p.2 Equation 5 is missing from the numbering
p.3 line 82 events instead of event
p.5 line 125 Error
Fig 4d: T written 2 time in the horizontal axis
Author Response
See attach document

Reviewer 2 Report
The brief communication is dedicated to the assessment of rockfall risk along roads and presents an example of how uncertainty can be managed in a simple way by assuming a given variability for most parameters. Hazard and risk probabilistic calculations are based on well-known and widely used formulations.
In general terms, the paper is very synthetic and a minor revision seems to the writer necessary to improve readability and understanding of all the steps of the proposed methodology. In particular:
- Some confusion exists about the parameter Pp. In the description of the conceptual formula for Risk calculation (lines 29-33), it is referred as "frequency of propagation for a given location"… What does it means? Is it referred to the rockfall propagation trajectories? How can it be evaluated as a function of volume? Moreover, is it assumed as the probability of impact with a vehicle (table 2)? What does it means "Integrated in the calculation from the integration of an order of magnitude of the volume" in table 2?
- Equations 8 and 9: what is the meaning of "flight"? Is it related to V(d) in table 2?
- What is D? It is referred as "road section width" (line 64) and as "Debris width" (table 2), please clarify.
- All the variables included in the risk calculation are assumed as statistical variables, with uniform distribution between a lower and a upper limit. It is not clear how such limits were defined and how the influence of volume can be included in the distribution. Anyway, this can be a strong simplification, a discussion on the influence of the frequency distribution of the parameters on the reliability of the results could be of value in the discussion section.
- All the data used in the paper were produced by a previous work by Hungr and co-authors. Many references are made to the latter but, in the opinion of the writer, some more details could be given to the reader in section 2 - Model data. For example, reference is made to the average value of N0 =100 events per year but it is not clear how many rockfall records are considered in the work and how long the period of observation is. Only the caption of Figure 1 reference is made to 390 events along 75 km of the road. Also, Figure 2 shows the values assumed for the probability of impact and the vulnerability as a function of volume. A brief description of the method used to obtain such values would be appreciated and a table reporting all the data and assumptions made in the previous work would improve the clarity of the paper.
- Although the observation period is not known, a large number of rockfall events is included in the data base. In many cases, this information is not available or incomplete (for example, no volume is associated to the rockfall event records). A brief discussion on the reliability of the results as a function of the reliability of input data could be included.
- In the discussion section, reference is made to several distribution functions that can be used to describe variables within the procedure. Again, a brief discussion could be included on the influence of the assumed distribution on the final results and its reliability.
In conclusion, the communication is a valuable example of how a rockfall analysis can be performed in probabilistic way at a small scale, i.e. with reference to large areas, where no extensive data collection can be done. It is therefore interesting for anyone involved in the assessment and management of rockfall risk along roads and it is certainly worth publishing within the Special Issue, after the minor revisions and the requests listed above will be addressed.
Author Response
See attach document
